# Molecular Alterations in Gastric Intestinal Metaplasia

**DOI:** 10.3390/ijms22115758

**Published:** 2021-05-28

**Authors:** Paulius Jonaitis, Limas Kupcinskas, Juozas Kupcinskas

**Affiliations:** Department of Gastroenterology, Lithuanian University of Health Sciences, 50161 Kaunas, Lithuania; limas.kupcinskas@lsmuni.lt (L.K.); Juozas.Kupcinskas@lsmuni.lt (J.K.)

**Keywords:** intestinal metaplasia, molecular alterations, genetic variations, micro-RNAs, microbiome, gastric cancer

## Abstract

Gastric cancer (GC) remains one of the most common causes of mortality worldwide. Intestinal metaplasia (IM) is one of the preneoplastic gastric lesions and is considered an essential predisposing factor in GC development. Here we present a review of recent most relevant papers to summarize major findings on the molecular alterations in gastric IM. The latest progress in novel diagnostic methods allows scientists to identify various types of molecular alterations in IM, such as polymorphisms in various genes, changes in the expression of micro-RNAs and long noncoding RNAs, and altered microbiome profiles. The results have shown that some of these alterations have strong associations with IM and a potential to be used for screening, treatment, and prognostic purposes; however, one of the most important limiting factors is the inhomogeneity of the studies. Therefore, further large-scale studies and clinical trials with standardized methods designed by multicenter consortiums are needed. As of today, various molecular alterations in IM could become a part of personalized medicine in the near future, which would help us deliver a personalized approach for each patient and identify those at risk of progression to GC.

## 1. Introduction

Despite its declining incidence over the past century [1], gastric cancer (GC) remains one of the most common cancers and causes of mortality worldwide. Based on the Global Cancer Observatory (GLOBOCAN) database by the World Health Organization (WHO) from the year 2020, GC ranks fifth in the incidence and fourth in the mortality list of various types of cancer [2]. Most gastric cancers are adenocarcinomas and have a growing number of various classification systems [3]. A widely used histological classification by Lauren stratifies GC adenocarcinomas into diffuse and intestinal types based on the glandular structure [4]. Intestinal type GC is the most common, and evidence shows that it is associated with gastric intestinal metaplasia [5]. Anatomically gastric cancer is classified into cardia GC and non-cardia GC (arising from other stomach parts) [1]. *Helicobacter pylori* (*H. pylori*; HP) infection causes chronic gastritis and is the biggest risk factor in the etiopathogenesis of non-cardia GC [6,7,8,9]. In the year 1994, *H. pylori* was classified as a Class I carcinogen by WHO [10] and remained the only bacterium given this classification [11]. Most *H. pylori* strains carry a virulence factor—cytotoxin-associated gene A (CagA), which encodes bacterial oncoprotein involved in gastric carcinogenesis [3,12]. Other well-known risk factors for GC are age, family history, smoking, alcohol consumption, high salt intake, diets low in fiber, fruit and vegetables, lack of physical activity, etc. Host-related factors, such as ABO blood type, genetic predisposition, and environmental factors, have been shown to contribute to gastric tumorigenesis as well [1,3].

The histological progression of gastric cancer, which is generally considered to be triggered by *H. pylori* infection [13], was first described by Correa et al. in 1975 and has been studied extensively ever since [14]. Therefore, it is known as the Correa’s cascade of gastric carcinogenesis (Figure 1.), which demonstrates histological pathway from normal gastric mucosa to gastric carcinoma (normal gastric mucosa → non-atrophic gastritis → atrophic gastritis → intestinal metaplasia → dysplasia → gastric cancer) [14,15]. Gastric intestinal metaplasia (IM) is one of the histopathological preneoplastic lesions of the stomach and is considered an essential predisposing factor in the development of intestinal-type GC [16,17]. Even though it is considered that intestinal metaplasia cannot be reversed [6], the possibility remains in the interest of scientists [18,19]. Gastric intestinal metaplasia has no specific treatment [20]. The progression of *H. pylori*-associated IM could be halted by the eradication of this bacterium. However, studies suggest that the regression of IM after *H. pylori* eradication is still highly questionable [21,22]. Regular surveillance in high-risk patients and the prevention of IM is one of the main management methods recommended by the guidelines. Modern treatment methods, such as endoscopic mucosal resection, are currently being investigated to treat gastric IM and show promising results [20]. Gastric IM is usually diagnosed by performing upper endoscopy with biopsy, and there are several histologic classification systems to determine the risk of IM progression to GC [23,24]. Nevertheless, scientists are still looking for other diagnostic approaches for premalignant gastric lesions, which could be used for screening purposes and to tailor proper treatment for each patient [25,26,27].

Modern diagnostic methods allow scientists to evaluate and examine each individual’s genetic and epigenetic information [28], and personalized medicine is expected to become one of the main diagnostics and therapeutic approaches in the near future [29,30]. Gastroenterology is a field of medicine that has robust feasibility to be involved in various types of molecular testing. Many potential novel biomarkers are to be used in the diagnosis of gastrointestinal (GI) diseases, including gastric carcinogenesis [31,32].

Much effort is being put into the research and systemization of gastric cancer; however, the focus on specific preneoplastic gastric lesions is still scattered. Therefore, this review article aims to summarize the findings of the most recent publications, studies, and information on the molecular alterations in gastric intestinal metaplasia, which are presented in Figure 2. 

## 2. Genetic Variations in IM

Various gene polymorphisms have been extensively studied as potential biomarkers for disease prediction, including premalignant gastric conditions and gastric cancer [33,34,35,36,37,38]. The most recent advancements in genomics enable scientists to identify single nucleotide polymorphisms (SNPs), which are the most frequent type of genetic variations in humans [39,40]. SNPs result from the alterations of a single nucleotide in a DNA strand [41,42]. SNPs in genes responsible for cell cycle regulation, metabolism, and repair processes are associated with increased susceptibility to various types of cancer [43,44].

It has been proposed that genetic variations in the nucleotide-binding oligomerization domain 1 (NOD1) receptors gene could be associated with premalignant gastric conditions, including intestinal metaplasia. A study from Turkey has shown that subjects with NOD1 796A/A and NOD1 796G/A genotypes had a significantly increased risk (compared to having wild-type NOD1 genotype) for antral IM, with odds ratios being 39.8 and 2.7, respectively [45]. However, another study found no significant associations between premalignant gastric conditions and polymorphisms not only of NOD1 but also the angiotensin-converting enzyme (ACE; rs4646994), toll-like receptor 4 (TLR4; rs11536889), and FAS/FASL (rs2234767, rs1800682, and rs763110) genes [35]. One of the most recent studies on TLR4 gene polymorphism by Nieuwenburg et al. has shown that the carriage of the minor C allele of the TLR4 gene was inversely associated with progression of IM (OR 0.6; 95 % CI 0.4–1.0) [46].

*H. pylori* infection and cytokine-mediated inflammatory responses have an important role in gastric cancer pathogenesis. Carriage of interleukin-1B-511 (IL-1B-511) T allele in *H. pylori* infected Chinese population was associated with an increased prevalence of IM (OR 2.0; 95% CI 1.0–3.7). The same study concluded that infection with HP vacA m1 genotype was also related to a significantly higher prevalence of IM (OR 1.8; 95 % CI 1.1–3.0), and the simultaneous presence of both host and virulence factors increased the risk of IM even further (OR 5.7; 95% CI 2.0–16) [47]. Another study also investigated the association between genetic polymorphisms in interleukin genes and premalignant gastric conditions in HP infected population. An elevated risk of intestinal metaplasia (OR = 5.58, 95 % CI 3.86–8.05) was associated with HP positive subjects, carrying the interleukin-22 (IL-22) rs1179251 CC genotype. Interestingly, the same study calculated decreased odds ratios for IM in subjects who carried the CG or GG genotype of the already mentioned Il-22 gene (OR = 0.24; 95 % CI 0.16–0.36 and OR = 0.32; 95 % CI 0.15–0.67, respectively). What is more, the interactions between various polymorphisms of cytokine genes and *H. pylori* infection were also analyzed. The study concluded that HP-positive people carrying the A allele of interleukin-32 (IL-32) rs2015620 gene (AA or AT) had an increased risk of GC and precancerous gastric lesions, including IM (OR = 9.08; 95 % CI 4.88–16.91) [48]. This study has shown significant interactions between the SNPs of cytokine genes and *H. pylori* infection, which suggests that genetic and environmental factors play an important role in developing precancerous gastric lesions.

Serum pepsinogen is one of the biomarkers proposed for screening purposes of premalignant gastric conditions and GC [49,50,51]. It was concluded in a study by Con et al. that alongside low (L-PG) or very low (VL-PG) pepsinogen levels and CagA positive *H. pylori* infection, the presence of IL-1B+3954T and IL-1 receptor antagonist homozygous 2 allele (IL*2/2) genetic variants were also associated with an increased risk of gastric IM as single markers. The results were debatable because the presence of CagA antibodies and IL-1B+3954T allele showed good sensitivity (96% and 80% respectively) and very low specificity (47% and 49%, respectively). The detection of VL-PG levels and had a mediocre sensitivity (78%) and specificity (79%), whereas the carriage of IL*2/2 genotype showed good specificity (89%) and low sensitivity (40%). The combination of VL-PG, CagA antibodies, IL-1β+3954T, and IL*2/2 showed an even further increased risk for IM (OR 30.9) with a sensitivity of 99.5% and specificity of only 17.4% [52].

Research by Zabaleta et al. analyzed 22 SNPs within eight genes (IL1B, IL8, IL6, TNF, PTGS2, ARG1, IL10, and TGFB1) and compared the findings between Caucasians and African-Americans. Even though none of the mentioned polymorphisms were associated with premalignant gastric conditions in Caucasians, the haplotype IL1B−511T/−31C/+3954T carried by African-Americans showed an increased risk of gastric intestinal metaplasia or dysplasia (OR 2.51; 95% CI 1.1–5.5) when compared to the most common haplotype T-C-C. The presence of allele T at the IL1B+3954 position was the only difference between the two compared genotypes [53].

Another study concluded that interleukin 10 (IL10) -1082 A allele showed a marginally increased risk of IM (OR = 1.43, 95% CI: 0.96–2.13), and the researchers found a 60% increased risk of IM and dysplasia combined (OR 1.62, 95% CI: 1.10–2.38) in the subjects carrying the already mentioned genetic variations [54].

A Chinese study investigated the associations between the polymorphisms of interleukin-8 (IL-8), macrophage migration inhibitory factor (MIF) genes, and premalignant gastric conditions in a high-risk area for gastric cancer. An increased risk of IM was found in subjects carrying IL-8-251 AT genotype (OR = 2.27; 95% CI 1.25–4.14), IL-8-251 A allele (OR = 2.07; 95% CI 1.16–3.69), MIF-173 CC genotype (OR = 2.27; 95% CI 1.16–4.46) and MIF-173 C allele (OR = 3.84; 95% CI 1.58–9.34). When accompanied by *H. pylori* infection, the risk for IM in subjects carrying the MIF-173 C allele was elevated significantly as well (OR = 2.93, 95% CI 1.28–6.60) [55]. Most of the previously mentioned SNPs in cytokine genes and their connections to GC and various precancerous gastric lesions were summarized by Negovan et al. in 2019 [56]. 

It has been proposed that the expression profiles of various mRNAs are altered in normal gastric tissue compared to gastric cancer mucosa. Several mRNAs have been suggested as novel biomarkers in the diagnostics of GC [57,58,59]. The caudal-type homeobox genes 1 (CDX1) and 2 (CDX2) have been identified to play an important role in the development of gastrointestinal cancers as well as precancerous gastric lesions [60]. A Japanese study concluded that CDX1 and CDX1 mRNAs were highly abundant in the intestinal mucosa of healthy individuals and none of these mRNAs were found in normal gastric mucosa. However, both CDX1 and CDX2 mRNAs were detected in the mild and severe types of IM, suggesting that these mRNAs could be involved in intestinal phenotypic differentiation [61]. Another study performed the analysis of DNA methylation in gastric cardiac mucosa. It revealed that apart from promoter hypermethylation, a reduced mRNA expression of the homeobox A5 (HOXA5) gene (which functions as a tumor suppressor in the development of GC) was also observed in gastric cardiac IM, suggesting a possible role of this gene in the development of gastric IM [62]. A Brazilian study investigated the expression of telomerase reverse transcriptase (TERT) gene mRNA in normal gastric mucosa, precancerous gastric lesions, and gastric cancer. The study concluded that progressively increased TERT mRNA expression levels were observed in IM and GC groups which suggests that this gene could be involved in gastric carcinogenesis [63]. Altered expression of cytokine genes mRNAs have been associated with premalignant gastric lesions, and higher expression of CYP1A2, CYP3A4, and CYP2D6 mRNAs were observed in intestinal metaplasia patients [64]. 

One of the most significant limitations for the clinical use of SNPs is the fact that most of the significant associations between numerous genetic variations and gastric intestinal metaplasia have an odds ratio in the range of 1.2–2.0. Therefore, their ability to be used for diagnostic purposes remains very limited. More advanced research of genetic variations is expected to come in the near future, and some of the most modern diagnostic approaches, such as genetic risk scores (GRS) or whole exome sequencing (WES) studies, are yet still to be applied in the diagnosis and prognosis of premalignant gastric conditions, including intestinal metaplasia.

## 3. MicroRNAs in IM

Noncoding RNAs (ncRNAs) are RNA molecules, which are not translated into proteins [65]. Even though they lack protein-coding potential, ncRNAs play an important role in many physiological functions and modulate complex molecular and cellular processes [66,67]. Moreover, they have been identified as tumor suppressors or oncogenic drivers in various types of cancer [67,68,69]. Examples of ncRNAs are microRNAs (miRNAs), long noncoding RNAs (lncRNAs), transfer RNAs (tRNAs), ribosomal RNAs (rRNAs), circular RNAs (cRNAs), and others [65,70]. A growing number of functional studies suggest that miRNAs are involved in different stages of gastric carcinogenesis [71,72]. It has even been proposed that the regulation of miRNA expression could be a novel strategy in the chemoprevention of human gastric malignancy [73].

An increase in the relative expression levels of miRNA-146a and miRNA-155 (7.5- and 124.5-fold respectively) was observed in *H. pylori* positive patients with intestinal metaplasia compared to healthy individuals [74]. Another study investigated bile regurgitation-induced gastric IM. It was reported that the upregulation of miR-92a-1-5p was observed in these patients, which led to a suppression of its target FOXD1 and high CDX2 (which, as previously mentioned, is associated with intestinal phenotypic differentiation) levels in IM tissues. Therefore, it was concluded that the suppression of miR-92a-1-5p could be one way to prevent IM in the cases of bile acids reflux [75].

A Chinese study investigated the miRNA17-92 cluster, which has seven members (miR-17-5p, miR-17-3p, miR-20a, miR-18a, miR-92a, miR-19a, and miR-19b) because, according to the literature, these miRNAs were associated with various types of cancer. It was reported that the expression of the miR-17-92 cluster members was up-regulated not only in gastric cancer patients but also in intestinal metaplasia subjects. Interestingly, miRNA17-92 levels were even statistically higher in IM patients compared to GC group. Furthermore, the expression of all the miR-17-92 members, especially miR-20a-5p with the largest area under the curve (0.996), highest sensitivity (98%), and sensitivity (95%), had diagnostic values for distinguishing intestinal metaplasia patients from healthy subjects or gastric cancer patients [76]. 

A Chinese study by Zhu et al. investigated the relationship between *H. pylori* CagA protein and miRNAs. The research concluded that the expression of miRNA-584 and miRNA-1290 was up-regulated in CagA-transformed gastric cells. Overexpression of these miRNAs caused intestinal metaplasia of gastric epithelial cells in knock-in mice. Therefore, it was concluded that *H. pylori* Cag A protein induces this miRNA pathway and interferes with cell differentiation [77].

Some miRNAs are down-regulated in the gastric carcinogenesis process, including intestinal metaplasia. It was reported that miR-490-3p levels decreased gradually in the progressive changes from gastritis to IM to HP-negative and were the lowest in HP-positive gastric cancer subjects [78]. Similar downregulation of miR-490-3p was observed in another Chinese study, in which *H. pylori* and N-methyl-N-nitrosourea (MNU)-induced gastric carcinogenesis was investigated [79]. Gastric intestinal metaplasia was also associated with the down-regulation of miR-30a [80].

In conclusion, we can see that the published studies have shown possible alterations of miRNAs expression in gastric IM. However, these findings are still hardly applicable in real-life clinical practice, and all of the results need to be confirmed in prospective cohort studies. Further progress in modern diagnostic and treatment methods could provide a potential possibility to regulate various miRNAs and, as a result, have an impact on the gastric carcinogenesis process.

## 4. Long-Non-Coding RNAs in IM

There is a clear lack of studies, including lncRNAs, one of the subtypes of noncoding RNAs, and preneoplastic gastric lesions. It was observed that the expression level of lncRNA GCAWKR increased from normal gastric tissue to intestinal metaplasia (IM), dysplasia, and GC [81]. Genome-wide analysis of lncRNA profile in human gastric epithelial cells has shown that the expression of lncRNAs XLOC_004122 and XLOC_014388 was decreased in HP-positive IM patients [82]. One of the Chinese studies investigated four candidate lncRNAs in the gastric carcinogenesis process. Even though three of the four lncRNAs had no associations with IM and gastric carcinogenesis, the levels of one lncRNA–BC041951, which was later named gastric cancer–associated lncRNA1 (GClnc1), were gradually increasing from normal gastric mucosa to IM, to dysplasia, to gastric cancer and was associated with poor prognosis in GC patients [83].

## 5. Microbiome Alterations in IM

Different profiles of the microbiome have been identified in most parts of the human body. However, the richest microbiome is found in the gastrointestinal tract, and its alterations have been associated with various diseases [84,85,86]. The gut microbiome plays an important role in maintaining a mucosal immune response, and dysbiosis could lead to gastric inflammation [87,88,89]. It has been reported that *H. pylori* infection alters gastric microbiome structure [90,91,92,93]. *H. pylori* is the most important gastric microbiome member, and it displays the highest relative abundance when present, but the stomach has a diverse microbiota when it is absent. The most abundant phyla in HP-positive and HP-negative patients are *Proteobacteria*, *Firmicutes*, *Actinobacteria*, *Bacteroidetes*, and *Fusobacteria* [94]. The microbiota changes gradually in the process of gastric carcinogenesis [95,96]. Therefore, it could be used as one of the biomarkers to determine premalignant gastric lesions, including IM.

A study with a very small cohort (10 subjects with IM and 10 healthy controls) investigated the gastric and duodenal microbiome differences between the two groups. The study concluded that the diversity of the duodenal microbiota in the control group was higher than that of IM group. Moreover, a significant difference in the duodenal microbiota structure was observed between the two groups. *Lactococcus*, *Flavobacterium*, *Psychrobacter*, *Mysroides*, *Enhydrobacter*, *Streptococcus,* and *Leuconostoc* were enriched in patients with IM. Interestingly, the status of *H. pylori* infection did not influence the structure of duodenal microbiota. Still, a statistically significant difference in gastric mucosal microbiota structure was observed between HP-positive and HP-negative patients in both groups [97].

Another study observed a decreasing gastric microbiota diversity (ranging from 8 to 57 bacterial genus) trend from non-atrophic gastritis (NAG) to intestinal metaplasia to gastric cancer (statistically significant difference was observed between NAG and GC groups). Moreover, the most abundant phyla in all three groups were Firmicutes (*Lachnospiraceae* and *Streptococcaceae* representing over 20%) and Proteobacteria, representing almost 70% of all phyla. The study showed five taxa with a decreasing trend from NAG to IM to GC (two *Saccharibacteria*, two *Porphyromonas,* and one *Neisseria*). Meanwhile, two taxa increased from NAG to IM to GC (*Lactobacillus* and *Lachnospiraceae*) [98].

A study by Sung et al. analyzed the alteration of the gastric microbiome after *H. pylori* eradication. Firstly, a significant reduction of gastric mucosal inflammation was observed one year after *H. pylori* eradication with OAC (omeprazole, amoxicillin, and clarithromycin) regimen compared to the placebo group. Even though IM changes were similar between the OAC-treated and placebo groups, the enrichment of *Peptostreptococcus* and depletion of *Lachnospira* in patients with IM before OAC treatment was observed. Among subjects with IM at baseline, *Pseudomonas*, *Peptostreptococcus*, *Halomonas,* and *Parvimonas* were enriched in those with progressed or persisted IM, and *Peptostreptococcus* was consistently positively associated with IM, before and after OAC treatment. It was observed that different sets of bacteria were associated with IM before and after *H. pylori* eradication. Among subjects with no IM at baseline, *Mesorhizobium* and *Cupriavidus* were enriched while *Actinomyces* were depleted in those with IM following *H. pylori* eradication [99].

Interestingly, research by Korean scientists showed that the evenness and diversity of gastric microbiota in GC group were increased compared to chronic gastritis (CG) and intestinal metaplasia groups. The GC group showed a significant decrease in the relative abundances of *Epsilonproteobacteria* class compared to the CG and IM groups. In contrast, the relative abundance of *Helicobacteraceae* family was significantly higher in the CG and IM groups [100].

Another Korean study analyzed the gastric microbiome of 138 patients based on the disease (CG, IM, and GC) and *H. pylori* infection status. Cyanobacteria were significantly reduced in the *H. pylori* negative IM group (4 %) compared to the *H. pylori* negative CG group (14%). Moreover, a statistically significant relative abundance of Rhizobiales was observed in the *H. pylori* negative IM group (15%) compared to the HP negative CG and HP negative GC groups (2% and 3 %, respectively). Interestingly, the study concluded that after a successful *H. pylori* eradication, the gastric microbiome resembled that in the HP-negative IM group [101].

Another study also investigated the changes of the gastric microbiome in the progression from healthy individuals to gastric cancer. The results revealed that the microbiota profile of intestinal metaplasia and chronic gastritis patients was quite similar—Acidobacteria, Gemmatimonadetes, Proteobacteria, and Verrucomicrobia were enriched in these groups. At the genus level, the abundance of *Halomonas*, *Shewanella*, *Aquincola,* and *Sphingomonas* was significantly higher in IM and CG groups. After analyzing each disease stage separately, the authors concluded that IM was featured by a higher abundance of *Aquincola tertiaricarbonis* and a taxon of *Sphingomonas* genus. What is even more important, the study investigated the possibility of using gastric microbiota as a predictive indicator for stages of gastric carcinogenesis. The constructed predictive model was able to identify the gastric histological type accurately. However, the calculated error rate was higher in CG and IM groups compared to the other disease stages, which further indicated microbiome similarity between these two groups [102].

A Chinese study investigated the associations between *H. pylori* status, its eradication, fecal microbiota and *H. pylori*-related gastric lesions, including IM. It was found that the abundance of Bacteroidetes was significantly higher in the *H. pylori* negative group compared to past HP infection group (66% compared to 33% respectively). The same trend was observed when comparing normal gastric mucosa to intestinal metaplasia group (76% compared to 47% respectively). It was also observed that the abundance of Firmicutes and Proteobacteria was significantly higher not only in the IM group (32% and 20%, respectively) compared to normal mucosa (18% and 5% respectively) but also in the past *H. pylori* infection group compared to HP-negative group. The findings suggest that alterations of fecal microbiota, including the previously mentioned phyla, may be involved in *H. pylori* related gastric lesion progression [103].

In conclusion, most of the previously mentioned studies identified microbiome alterations in IM that may be involved in gastric tumorigenesis. They laid a foundation for future studies using bacterial markers for diagnosing premalignant gastric conditions, including intestinal metaplasia. However, larger-scale studies and significant efforts are still needed to standardize the process of microbiome analysis and the interpretation of the result to apply these findings to clinical practice.

## 6. Conclusions

In conclusion, the latest progress in novel diagnostic methods allows scientists to identify various molecular alterations in gastric intestinal metaplasia, such as polymorphisms in various genes, changes in the expression of miRNAs and lncRNAs, and altered microbiome profile. Some of these alterations have strong associations with intestinal metaplasia and a potential to be used as one of the tools for the screening, treatment, and prognostic purposes; however, the use of current findings in real-life clinical practice is still very limited. One of the most important limiting factors is the inhomogeneity of the studies, such as the lack of data on concomitant *H. pylori* infection, lack of risk modification by HP CagA status, heterogeneous methods used to assess preneoplastic gastric lesions, uneven or small cohorts, etc. Therefore, further large-scale studies and clinical trials with standardized methods designed by multicenter consortiums are needed. As of today, various molecular alterations in intestinal metaplasia could become a part of personalized medicine, which would help us deliver a personalized approach for each patient and identify those at risk of progression to gastric cancer.

## Figures and Tables

**Figure 1 ijms-22-05758-f001:**
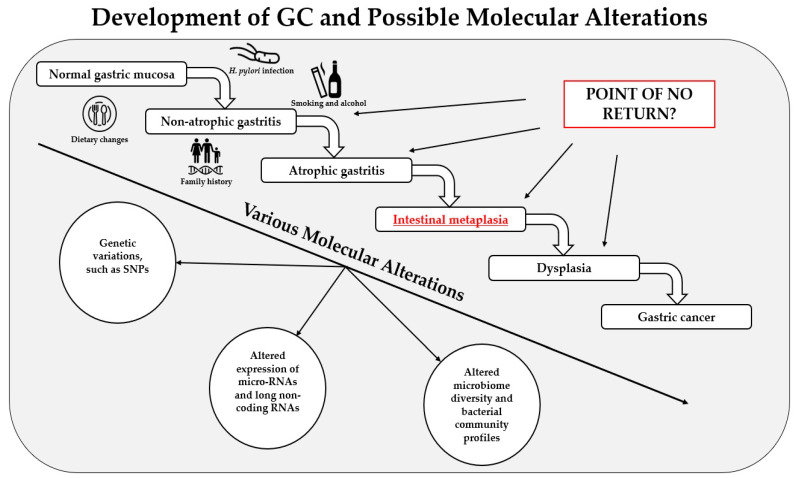
Development of gastric cancer (GC), the main causes and risk factors for it, and possible molecular alterations.

**Figure 2 ijms-22-05758-f002:**
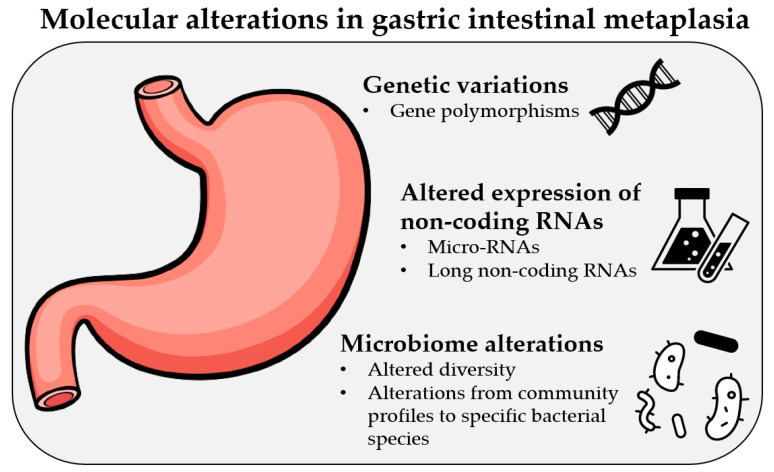
Most frequent molecular alterations observed in gastric intestinal metaplasia.

## Data Availability

Data sharing not applicable.

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
