# Peer review of "Molecular Alterations in Gastric Intestinal Metaplasia"

_ijms, 2021, doi:10.3390/ijms22115758_

Round 1

Reviewer 1 Report

This is an interesting and important review article discussing recent developments on genetic polymorphisms, altered micro- and lnc-RNA expression and the microbiome in Intestinal Metaplasia with a view to identifying those more at risk of developing gastric cancer.  

One key area that is neglected in the review is a dedicated section on recent developments on Altered expression of mRNA and/or protein in IM. Changes in CDX mRNA expression are mentioned in the “Genetic variations in IM” section but this should be expanded to include other recent studies comparing mRNA and/or protein expression in normal and IM tissue.

Minor comments: 

Describe "non-cardia GC" and "intestinal type GC" to those not familiar or new to the research area

Author Response

Response to Reviewer 1 Comments

Dear Reviewer,

We would kindly like to thank You for the revision of our manuscript and all of the comments. Below You will find our responses to each individual comment.

Point 1: One key area that is neglected in the review is a dedicated section on recent developments on Altered expression of mRNA and/or protein in IM. Changes in CDX mRNA expression are mentioned in the “Genetic variations in IM” section but this should be expanded to include other recent studies comparing mRNA and/or protein expression in normal and IM tissue.

Response 1: A paragraph (lines 157-159 and 166-177) describing other studies on altered mRNAs expression in gastric intestinal metaplasia has been added at the end of the section “Genetic variations in IM”. There is still a clear lack of studies on the expression of mRNAs in intestinal metaplasia; therefore only a few other studies have been described.

Point 2: Describe "non-cardia GC" and "intestinal type GC" to those not familiar or new to the research area.

Response 2: A short description of the main histological (diffuse and intestinal type GC) and anatomical (cardia and non-cardia GC) classification of gastric cancer has been added at the beginning of “Introduction” section (lines 28-34).

We hope that our changes and comments are suffice and the manuscript can be published. Thank You once again for Your time.

Regards,
Authors

Reviewer 2 Report

Authors should put appropriate figures in order to understand the data which you cited in section 3,4 and 5.

Author Response

Response to Reviewer 2 Comments

Dear Reviewer,

We would kindly like to thank You for the revision of our manuscript and all of the comments. Below You will find our responses to Your comment.

Point 1: Authors should put appropriate figures in order to understand the data which you cited in section 3,4 and 5.

Response 1: We have created a new figure (Figure 1) which demonstrates the process of gastric carcinogenesis, the main causes and risk factors for it and the possible molecular alterations which are described in sections 3, 4 and 5. In our opinion, a table which could summarize the main findings described in sections 3, 4 and 5 would be complex and, as a result, hard to read and would not help the reader to better understand the data described in the main body of the manuscript; therefore we decided not to add any tables and created a simple and easy-to-understand Figure 1.

We hope that our changes and comments are suffice and the manuscript can be published. Thank You once again for Your time.

Regards,
Authors

Reviewer 3 Report

This manuscript summarizes the findings of the most recent publications, studies and information on the molecular alterations in gastric intestinal metaplasia (genetic variations, microRNAs, long-non-coding RNAs, and microbiome alterations). In general, the manuscript is well-written and interesting. This reviewer has only a few comments to improve the paper. 1. Illustration or picture may be added to understand the lesion image of gastric intestinal metaplasia. 2. Clinical treatments for gastric intestinal metaplasia should be described.

Author Response

Response to Reviewer 3 Comments

Dear Reviewer,

We would kindly like to thank You for the revision of our manuscript and all of the comments. Below You will find our responses to Your comment.

Point 1: Illustration or picture may be added to understand the lesion image of gastric intestinal metaplasia.

Response 1: We have created a new figure (Figure 1) which demonstrates the process of gastric carcinogenesis, the main causes and risk factors for it and the possible molecular alterations which are described in the manuscript.

Point 2: Clinical treatments for gastric intestinal metaplasia should be described.

Response 2: A short part on the treatment of gastric intestinal metaplasia has been added in the “Introduction” section (lines 53-60).

We hope that our changes and comments are suffice and the manuscript can be published. Thank You once again for Your time.

Regards,
Authors

Round 2

Reviewer 1 Report

The authors have addressed the reviewers comments appropriately